# Computational Fluid Dynamics Analysis of Varied Cross-Sectional Areas in Sleep Apnea Individuals across Diverse Situations

W. M. Faizal *, C. Y. Khor, Suhaimi Shahrin, M. H. M. Hazwan, M. Ahmad, M. N. Misbah and A. H. M. Haidiezul

Faculty of Mechanical Engineering & Technology, University Malaysia Perlis, 02600 Arau, Perlis, Malaysia; cykhor@unimap.edu.my (C.Y.K.); suhaimishahrin@unimap.edu.my (S.S.); hazwanhanid@unimap.edu.my (M.H.M.H.); masniezam@unimap.edu.my (M.A.); mnur@unimap.edu.my (M.N.M.); haidiezul@unimap.edu.my (A.H.M.H.)
* Correspondence: wanmohd@unimap.edu.my; Tel.: +60-124-524-518

**Abstract:** Obstructive sleep apnea (OSA) is a common medical condition that impacts a significant portion of the population. To better understand this condition, research has been conducted on inhaling and exhaling breathing airflow parameters in patients with obstructive sleep apnea. A steady-state Reynolds-averaged Navier–Stokes (RANS) approach and an SST turbulence model have been utilized to simulate the upper airway airflow. A 3D airway model has been created using advanced software such as the Materialize Interactive Medical Image Control System (MIMICS) and ANSYS. The aim of the research was to fill this gap by conducting a detailed computational fluid dynamics (CFD) analysis to investigate the influence of cross-sectional areas on airflow characteristics during inhale and exhale breathing in OSA patients. The lack of detailed understanding of how the cross-sectional area of the airways affects OSA patients and the airflow dynamics in the upper airway is the primary problem addressed by this research. The simulations revealed that the cross-sectional area of the airway has a notable impact on velocity, Reynolds number, and turbulent kinetic energy (TKE). TKE, which measures turbulence flow in different breathing scenarios among patients, could potentially be utilized to assess the severity of obstructive sleep apnea (OSA). This research found a vital correlation between maximum pharyngeal turbulent kinetic energy (TKE) and cross-sectional areas in OSA patients, with a variance of 29.47%. Reduced cross-sectional area may result in a significant TKE rise of roughly 10.28% during inspiration and 10.18% during expiration.

**Keywords:** CFD; breathing disorders; turbulent kinetic energy; obstructive sleep apnea; real patient

## 1. Introduction

Obstructive sleep apnea (OSA) substantially impacts the sleep quality of affected individuals. In recent times, scholars have directed their attention toward exploring the influence of body characteristics on this condition and its effects on individuals with OSA. Their studies have shed light on the importance of considering various body characteristics in comprehending the phenomenon of OSA. Wang et al. [1] discovered that the weight of a patient contributes significantly to the occurrence and impact of OSA. These findings were the same as those of Tham [2], who demonstrated that obesity significantly contributes to the onset and development of obstructive sleep apnea (OSA). Nonetheless, notable enhancements in managing obstructive sleep apnea (OSA) can be achieved by promoting substantial weight loss within the range of approximately 7% to 11%. In the case of neonates, the underdeveloped nature of their nasal cavity significantly influences the configuration of their airways. More precisely, the lack of an inferior meatus in neonates hinders airflow through the inferior regions, resulting in an uneven distribution of airflow [3]. Modena et al. [4] discovered that even slight changes in hip-to-waist ratio, neck circumference, and body mass index (BMI) can impact the likelihood of developing obstructive sleep apnea syndrome

(OSAS). Higher values in these variables are associated with a greater risk of developing OSAS. While it is important to acknowledge the limitations in published studies, it has been shown that behavioral, pharmacological, and surgical interventions can facilitate weight loss, leading to improvements in the severity of OSAS, reversal of common comorbidities, and enhanced quality of life [5]. After weighing the evidence, feasibility, and acceptability of these interventions, the expert panel strongly advised implementing a comprehensive lifestyle intervention for individuals with obstructive sleep apnea (OSA) who are overweight or obese. This intervention should encompass (1) adoption of a low-calorie diet [6], (2) engagement in exercise or increased physical activity [7], and (3) receiving behavioral guidance [8]. This holistic approach aims to address various aspects of lifestyle to enhance overall well-being and manage the specific challenges associated with OSA in individuals with higher body weight. Managing weight loss is suggested through potential strategies such as increasing physical activity through exercise and adopting a reduced-calorie diet. However, implementing these approaches is contingent upon the individual's specific circumstances. In some instances, pharmacological therapy or bariatric surgery may be considered appropriate options for patients requiring additional assistance with weight loss [9]. In inpatient populations, the upper airway length tends to be longer. It exhibits a robust correlation with body weight. An increase in body weight results in significant fat infiltration within the tongue, inducing the downward movement of the hyoid. This situation, in turn, elongates the airway in affected individuals. The correlation between the apnea–hypopnea index (AHI) and the airway length, along with the size of the tongue, is prominently evident [10]. Unexpectedly, there are substantial distinctions between healthy males and patients in terms of both the distance parameter "h" and the angle near the occipital bone. These variations arise from the diverse backward tilt angles, and both parameters exhibit a robust correlation with the AHI. In patients, the downward movement of the hyoid bone and the tilted head contribute to 67.4–80.5 percent and 19.5–32.6 percent of the lengthening of the airway. Furthermore, there is a strong correlation between the size of the parapharyngeal fat pad and the AHI [11].

As a result, the cross-sectional area's information about the cavity and airways may affect the severity of OSA in patients. Faizal et al. [12] recognize turbulent kinetic energy (TKE) as a valuable tool for understanding various medical disorders, including OSA. Turbulent kinetic energy (TKE) was employed to assess the state of turbulent flow within the upper airway and to compute the energy balance of the airflow. This measurement provides insights into turbulence and energy distribution characteristics within the airway system. Elevated turbulent kinetic energy (TKE) within the upper airway can lead to tissue vibration during breathing, potentially triggering sleep apnea. Moreover, TKE has been used to explore the airflow dynamics of the upper airway and anatomy, contributing to the development of virtual reality surgery for sleep-disordered breathing (SDB) [13].

Furthermore, airway stenosis reduces the airways' cross-sectional area by narrowing them, hindering airflow into the lungs. Various diseases can cause airway stenosis [14], such as infection, a chronic inflammatory disease [15], trauma, and cancer. The stenosis airway correction is typically achieved by endoscopic or open surgery to expand the lumen. Various factors, including individual anatomy, larynx and trachea functionality, patient-related considerations, and the availability of facilities, play a role in influencing the surgical procedure. In cases of airway stenosis, utilizing a 50 percent helium–oxygen mixture during high-frequency jet ventilation results in an 18 percent increase in minute volume. This increase is achieved with airway pressures equal to or lower than those generated using 100 percent oxygen [16]. Simulation analysis allows for a better understanding of the airflow patterns in airways affected by stenosis, particularly concerning various disease conditions [17]. Velocity profile and pressure drops are the popular parameters used in the simulation analysis. The particle distribution during the inhale and exhale breathing can also be visualized in the numerical method.

Examining the flow dynamics in the upper airway offers valuable insights for addressing sleep apnea and respiratory illnesses. Employing CFD simulations and particle

image velocimetry experiments allows for visualizing the inhalation and exhalation flow patterns [18]. The "glottal jet" phenomenon is less apparent at higher flow rates, and the separation zone is smaller during inhalation. Nevertheless, the peak velocity observed was notably lower than the inhalation phase. The flow patterns in the tracheal region remained unchanged by the exhalation breathing modes [18]. Simulating the inhalation and exhalation patterns aids in visualizing and comprehending particle movement within the human airway. The nonstationary flow was represented by a steady flow, replacing the asymmetric cycle simulation. The volumetric flow rates encompassed maximum and average rates during inhalation [19]. Utilizing CFD simulation, with wall motion guided by magnetic resonance imaging, during both inhalation and exhalation offers novel perspectives on the authentic flow dynamics of breathing. The dynamic motion of the airway has the potential to generate physiologically realistic outcomes in the CFD simulation [20].

This study aims to explore turbulent airflow in the pharyngeal airway among individuals diagnosed with OSA, highlighting both inhalation and exhalation and examining the impact of cross-sectional areas in diverse OSA patients. To the best of the authors' knowledge, there is a notable absence of research concerning the influence of cross-sectional areas in various OSA patients. Hence, a notable research gap exists, especially in the simulation modeling analysis. Therefore, this study primarily explores body characteristics to comprehend the OSA phenomenon. However, it lacks the thorough investigation needed to elucidate the complexities of airflow, which is essential for a comprehensive understanding of OSA. The novelty of the research lies in its comprehensive airflow reaching the critical cross-sectional area. The TKE increase in the human upper airway (HUA) segment significantly surpassed other segments in the control scenario. The study revealed a strong correlation between maximum pharyngeal turbulent kinetic energy (TKE) and cross-sectional areas among patients with OSA.

## 2. Materials and Methods

In this investigation, a CFD simulation was performed to model the upper human airway, specifically customized to suit the characteristics of the study's subjects. The primary emphasis was on patients with OSA, considering the diverse cross-sectional areas of their upper airways. The objective was to examine the airflow patterns during both inhalation and exhalation. The three-dimensional HUA simulation models were created based on the CT-Scan data. ANSYS Fluent was utilized in the CFD simulation to create mesh elements, model the flow behavior, and define the boundary conditions. The grid sensitivity test was carried out to ensure that there was no grid effect on the simulation model. The solver applied in the simulation was also compared with that of the previous study. A detailed discussion of the methodology is presented in the following subsections.

### 2.1. Subject

The HUA's specific geometry was extracted using volunteer OSA patient subjects based on data availability, as detailed in Table 1. Written informed consent was provided by patients, and the study adhered to the principles of the Declaration of Helsinki regarding research involving human subjects. The research was approved by the Universiti Malaysia Perlis Research Ethics Committee, Perlis, Malaysia, with the reference UniMAP/PTNC(P&I)/JKETIKA (10/11/2021). As shown in Figure 1, 431 frames with a slice thickness of 0.3 mm were used to scan the subject's upper airway using an i-CAT Cone Beam 3D Dental Imaging System (version 3.1.62 supplied by Imaging Science International, Hatfield, PA, USA). The description of the airway boundary is established by applying a threshold based on the intensity of the gray image in the CT scan images. As detailed in previous research, these images were acquired while the patient was in an awake position, lying down and facing upward, as described in the previous work [21]. Each CT scan comprises 534 pixels × 534 pixels, with a pixel spacing of 0.3 mm × 0.3 mm. These CT scan images were stored in the Digital Imaging and Communications in Medicine (DICOM) format. Subsequently, these images were brought into the MIMICS software (version 15.0;

Materialise, Leuven, Belgium), a three-dimensional (3D) medical image processing tool. The Hounsfield-unit-based image segmentation of the hypopharyngeal airway (HUA) was conducted on the DICOM image series to delineate the 3D airway region. Following this, the 3D volume of the pharyngeal airway was generated and exported to facilitate the creation of the 3D model mesh.

**Table 1.** General characteristics of OSA patient.

|  | **Responder 1** | **Responder 2** |
|---|---|---|
| Sex | Female | Male |
| Age (years) | 33 | 44 |
| BMI (kg/m$^2$) | 28.5 | 25.42 |
| OSA Level | 12.5 (Mild) | 27.1 (Moderate) |

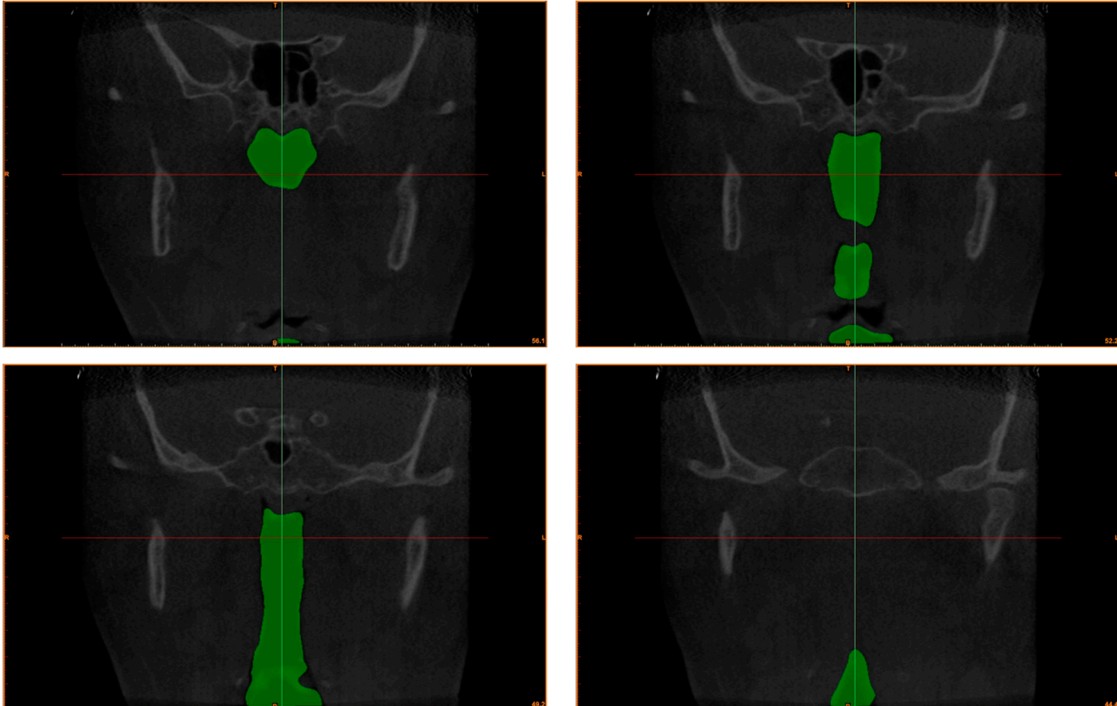

**Figure 1.** CT scan images of the OSA upper airway cavity.

The boundary conditions for the 3D model were established based on the model's surface, as indicated in Figure 2. The approach employed in this study integrates computational fluid dynamics (CFD) modeling with the segmentation of obtained computed tomography scans (CT scans). The CT scan image of the patient served as the foundation for creating the domain. The designated locations of P6 and P1 were employed to specify the inlet and outlet boundary conditions for inhalation to focus on the flow simulation. The model, initially created using MIMICS software, was then transferred to 3-Matic (version 15.0; Leuven, Belgium) to construct the airway's wall mesh. Afterwards, the preprocessor, ICEM (ANSYS Inc., Canonsburg, PA, USA), generated unstructured tetrahedral meshes for the airway volume. A similar meshing technique used in our previous work [12] was applied in the current study. The pharyngeal airway surfaces were meshed with a maximum face size of 2.0 mm. The mesh's size range spans from 2 mm to a minimum of 0.002 mm. Inflation was applied around the wall boundary of the pharyngeal airway mesh to enhance simulation accuracy. The y+ approach, which represents a dimensionless distance, was employed to estimate the initial thickness of the boundary layer. This estimation corresponds to a near-wall cell size of 0.272, providing a measure of the proximity of the computational grid to the physical wall in the simulation. With a skewness value

of 0.2142, indicating excellent cell quality within the recommended range of 0–0.25 by ANSYS, the mesh sensitivity grid study was conducted using various grid scales on the 3D airway model.

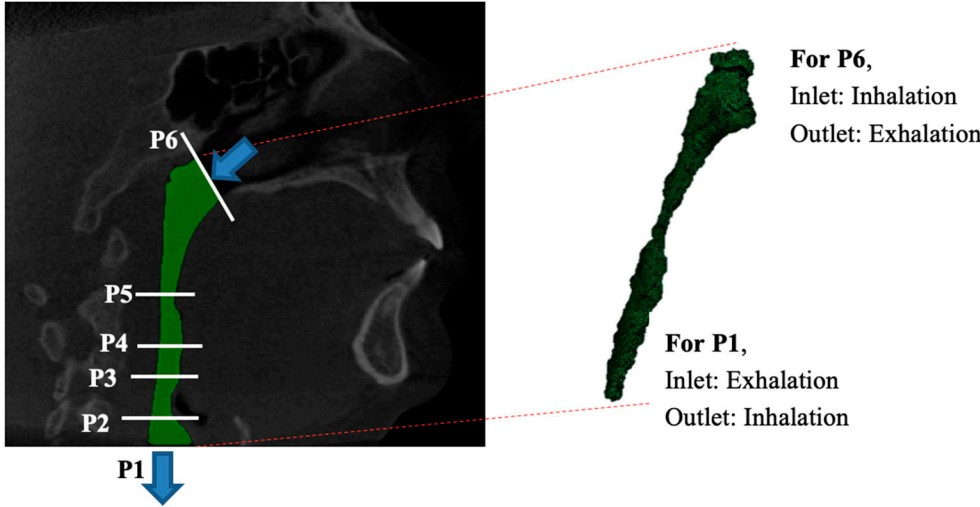

**Figure 2.** The lateral perspective of the meshed airway model of the upper airway. The CT scan's anatomically designated points, denoted as P1, P2, P3, P4, P5, and P6, correspond to specific locations: (P6) nasal choanae level; (P5) minimum cross-section area; (P4) tip of uvula; (P3 and P2) laryngopharynx; and (P1) base of epiglottis—representing the outlet of the computational fluid dynamics (CFD) model.

The steady-state turbulent model is the most established technique for modeling respiratory flows in the CFD analysis [22–24]. This study used ANSYS Fluent, which employs finite volume methods with fitted grids, to compute steady-state turbulent airflow in the airway. The assumption of a uniform velocity profile during breathing was made. The axial velocity was perpendicular to the flow inlet, specifically the nasopharynx [22,25]. Light breathing conditions were modeled at the inlet for both scenarios. The simulation incorporated a volume flow rate of 7.5 L/min [26]. The outlet was characterized by an average gauge pressure of 0 Pa, and a turbulent intensity of 10% was considered representative of the actual conditions [27].

Furthermore, the current study has limitations, such as considering the uniform breathing velocity. The airway geometry was scanned while the subject was awake and in an upright position. This study also excludes the sleep study, respiratory stage, and tongue control. The airway model was considered rigid in the simulation, and any influences stemming from soft tissue were ignored.

*2.2. Numerical Modeling*

The distinctive nature of the pharyngeal airway contributes to variations in airflow characteristics. Therefore, the simulation analysis employed a steady-state Reynolds-averaged Navier–Stokes (RANS) formulation, utilizing the k–ω shear stress transport (SST) turbulence model with low Reynolds number correction to compute the flow field in the pharyngeal airway for patients with obstructive sleep apnea (OSA). The popular k–ω SST turbulence model was considered in this work instead of the k–ϵ model due to the greater accuracy of the former in viscous near-wall region treatment while considering the effects of the adverse pressure gradient [28]. The SST model is recommended for high-accuracy boundary layer simulations, necessitating a very high grid resolution near the boundary [26]. The rigid model of the upper airway with a static wall was considered in the simulation without considering the fluid–structure interaction aspects. The assumption of a rigid model helps explore the effect of airway narrowing on a deformable airway's performance [28].

The simulation analysis considered and solved the continuity equation (Equation (1)) and momentum equation (Equation (2)) to describe the motion of incompressible airflow. However, a minimal temperature change was observed in the airway during this study. Thus, the energy equation and gravitational force were neglected.

$$\nabla \vec{v} = 0 \tag{1}$$

$$\rho u_i \frac{\partial u_j}{\partial x_i} = -\frac{\partial P}{\partial x_i} + \frac{\partial}{\partial x_i}\left[\mu\left(\frac{\partial u_i}{\partial x_j} + \frac{\partial u_j}{\partial x_i}\right) - \rho\overline{u_i' u_j'}\right] \tag{2}$$

where $u$ is the velocity, and $i$ and $j$ represents the Cartesian coordinates. An advection scheme employing a second-order upwind discretization scheme was selected. Convergence of results was deemed achieved when the residual level reached the predefined target of $1 \times 10^{-6}$.

### 2.3. TKE Model and Turbulent Reynolds Number

The Reynolds-averaged Navier–Stokes (RANS) equations describe the average flow field quantities over time in fluid dynamics, particularly turbulent flows. The turbulence closure problem in RANS involves introducing additional equations or models to account for the effects of turbulence.

$$u' = \sqrt{\frac{1}{N_i}\sum_{i=1}^{N}(u_i - \overline{u})^2} \tag{3}$$

where $N$ represents the number of samples in the signal incorporated in the ANSYS simulation, and the equation for the fluctuation component is equally applicable to the velocity signals v and w. With these fluctuation components ($u'$, $v'$, and $w'$), it is then possible to model the TKE as:

$$\frac{\partial(\rho k)}{\partial t} + \frac{\partial(\rho u_j k)}{\partial x_j} = \frac{\partial}{\partial x_j}\left[\left(\mu + \frac{\mu_t}{\sigma_k}\right)\frac{\partial k}{\partial x_j}\right] - \rho\varepsilon + P_k - \beta^*\rho k \tag{4}$$

with density ($\rho$); turbulent kinetic energy ($k$), also known as TKE; fluctuation velocity ($u_j$); viscosity ($\mu$); turbulent viscosity ($\mu_t$); turbulent Prandtl number for $k$ (($\sigma_k$); dissipation rate of turbulent kinetic energy ($\varepsilon$); production of turbulent kinetic energy ($P_k$); turbulent kinetic energy dissipation constant ($\beta^*$). By employing this equation, the velocity fluctuation utilized in the computation of TKE around the HUA can aid in assessing the severity of breathing blockage within the HUA. In comparison, kinetic energy (KE) is defined as:

$$KE = \frac{1}{2}\rho\left(u^2 + v^2 + w^2\right) \tag{5}$$

where $u$, $v$, and $w$ are the phase-averaged velocity. The definition of the turbulent Reynolds number, $Re_y$, is shown in Equation (6):

$$Re_y = \frac{Kinectic\ Energy}{Work\ of\ friction} = \frac{KE}{W} \tag{6}$$

The balance of *KE* is given in Equation (5), while the work friction force, *W*, is given in Equation (7):

$$W = \iiint_v \tau_{ij}dV \tag{7}$$

where $\tau_{ij}$ represents the total shear stress component, encompassing both molecular and turbulent friction, acting on the surface of the integration volume, *dV*.

### 2.4. Grid Sensitivity and Solver Verification

A grid sensitivity test has been conducted to assess grid sensitivity before advancing to solver verification, varying the total number of elements. Five numbers of elements ranging from 371 k to 3.29 million for the Responder 1 model and 383 k to 4.77 million for the Responder 2 model were compared for the grid sensitivity test. The simulation result converged at 1.29 million elements for the Responder 1 model with a deviation of 0.25% compared to 3.29 million elements and for Responder 2 converged at 1.8 million elements with a deviation of 0.31% compared to 4.77 million elements, as shown in Figure 3. Thus, the 3D simulation model with 1.29 million and 1.8 million elements for model Responders 1 and 2, respectively, was considered in this study. The dimensionless normal distance between the wall (or airway surface) and the closest mesh cell in the viscous sublayer is represented by the wall unit (y+). The y+ value was less than 1 for all individuals over the domain. This situation was enough to resolve the mesh model near wall flow dynamics for Responders 1 and 2, which are 0.04223 and 0.04831, respectively. After this, the research underwent solver verification, a process detailed in our prior publication [29], as shown in Figure 4. Using the solver (RANS $k-\omega$ SST), the current results showed an almost similar trend to the previous experimental and simulation results as mentioned in Figure 4. Thus, the results indicated that the RANS $k-\omega$ SST solver is validated for the current study, and the prediction using this solver is reliable.

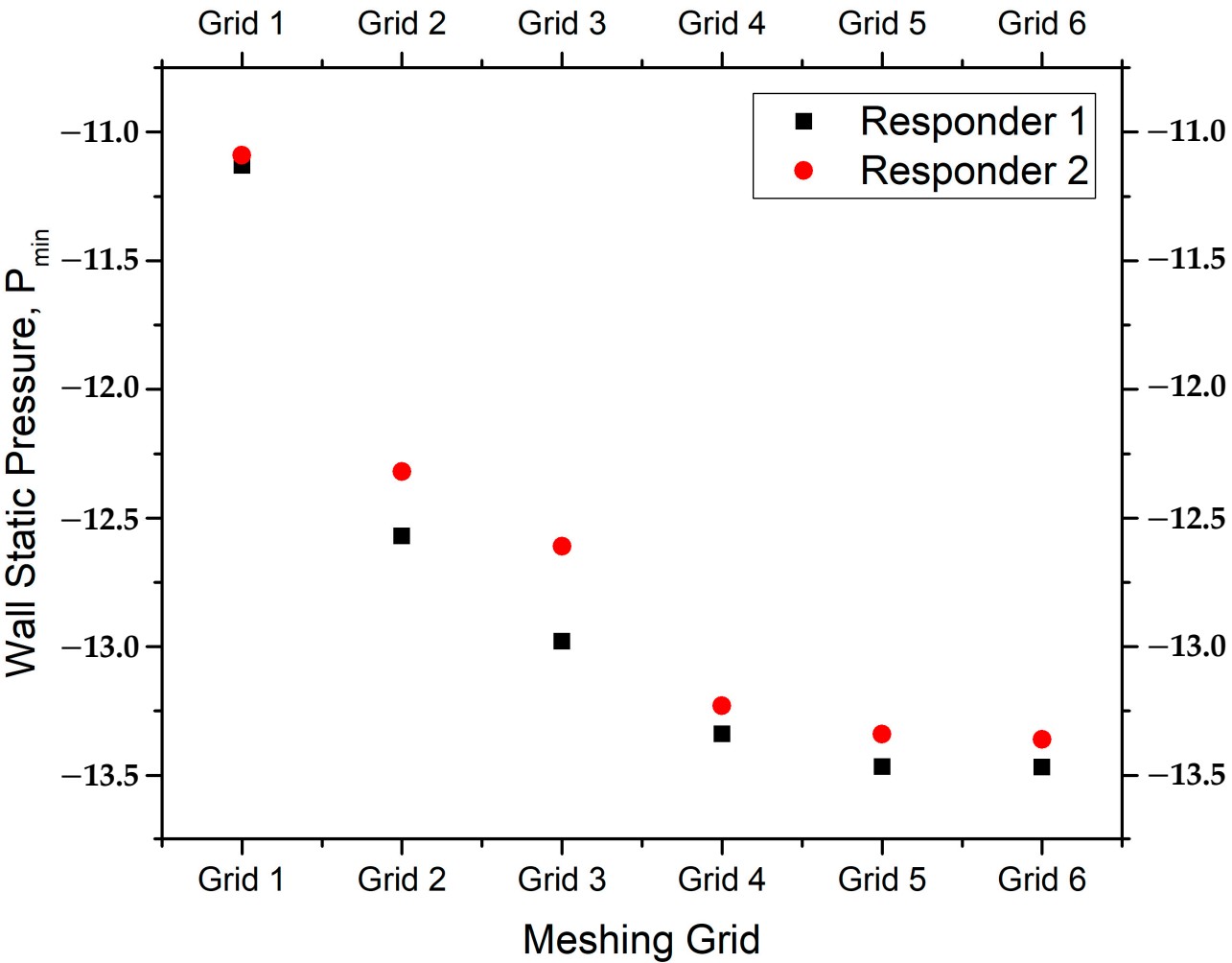

**Figure 3.** The grid sensitivity test on the various grid sizes for Responders 1 and 2.

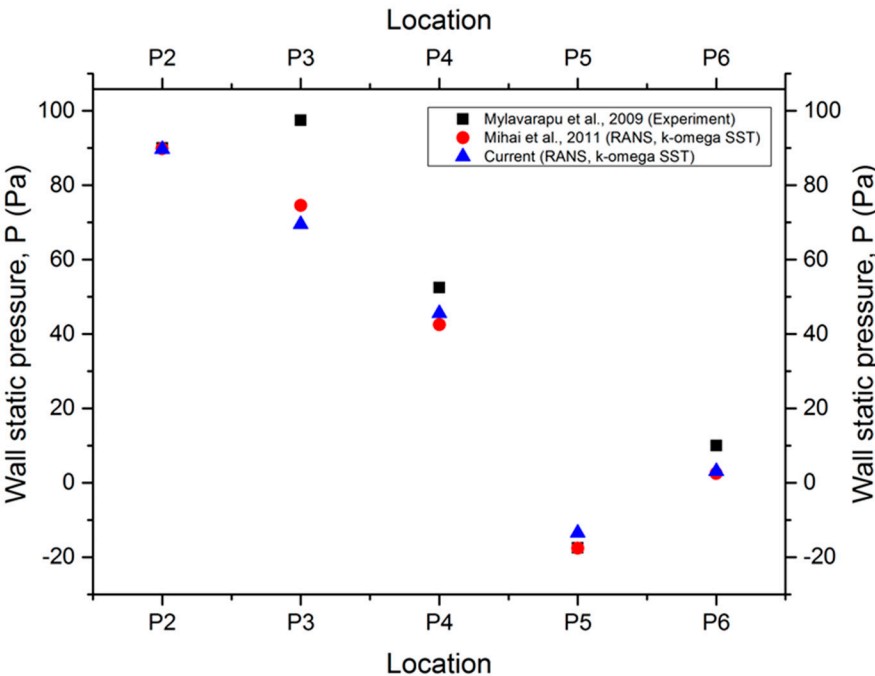

**Figure 4.** Solver verification with previous simulation and experimental results by Mylavarapu et al. [30] and Mihai et al. [31].

## 3. Results and Discussion

### 3.1. Cross-Sectional Area

In Figure 5, the hypopharyngeal airway (HUA) cross-sectional area changes are depicted across thirty-four planes in the z-direction (i.e., flow direction). The representation highlights that for Responder 1, the patient exhibits a minimum cross-sectional area of 7.68 mm$^2$ at a distance of 28 mm, while for Responder 2 it is 5.42 mm$^2$ at a distance of 31 mm from the inlet. The different cross-sectional areas between Responders 1 and 2 are around 29%. Figure 5 shows that both responders have different cross-sectional areas of the upper airway within 25–35 mm, which may be attributed to the individual features of the OSA patient's airway. Thus, the current study is significant in providing the airflow pattern to help medical practitioners individualize the OSA treatment based on underlying mechanisms. The narrowing of the cross-sectional area impacts the fluidity of the airflow as it traverses through the hypopharyngeal airway (HUA). TKE was used to explain this study's significant result to reveal the flow characteristics of the HUA because of the cross-sectional area.

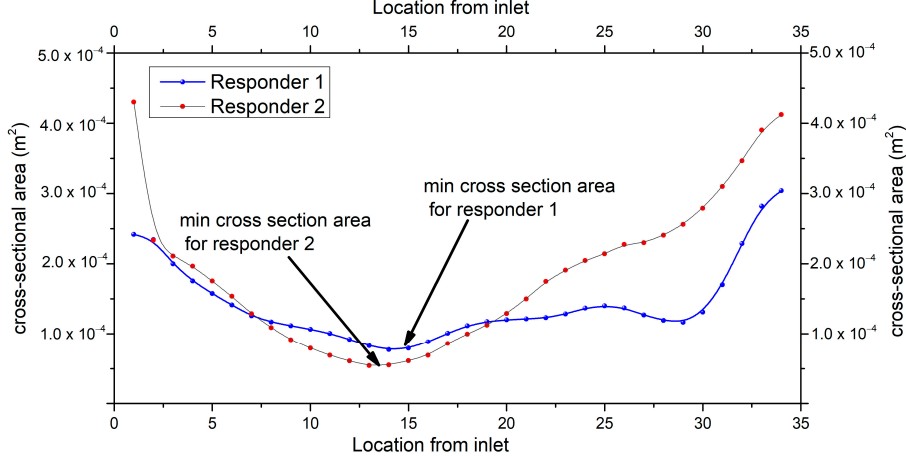

**Figure 5.** The generation of the cross-sectional area along the airflow direction in the airway model.

Moreover, Zhao et al. [32] demonstrated the significance of correlation analysis by revealing that pharyngeal volume and upper airway cross-sectional area were useful markers for screening sleep apnea patients. The discoveries by Lin et al. [33] endorse the notion that alterations in the cross-sectional configuration of the upper airway exert a substantial influence on airway resistance. The modification in the shape of the upper airway's cross-section can offer crucial clinical insights into an individual's patterns of upper airway collapse. Ryu et al. [34] recently completed an airway simulation to examine geometrical implications. Based on the computed results, they developed a model to anticipate the flow characteristics that cause contraction of the upper airway in sleep apnea patients.

### 3.2. Static Pressure

Figure 6 illustrates the static pressure (unit: Pa) exerted on the airway wall throughout the inhalation and exhalation phases, featuring distinct responders (i.e., cross-sectional areas). The simulation results discovered the maximum axial velocity and minimum static pressure in the small cross-sectional regions [27]. In terms of mechanism, the genioglossus is influenced by a locally medicated working effective reflex system that responds to negative pharyngeal pressure and central respiratory drive.

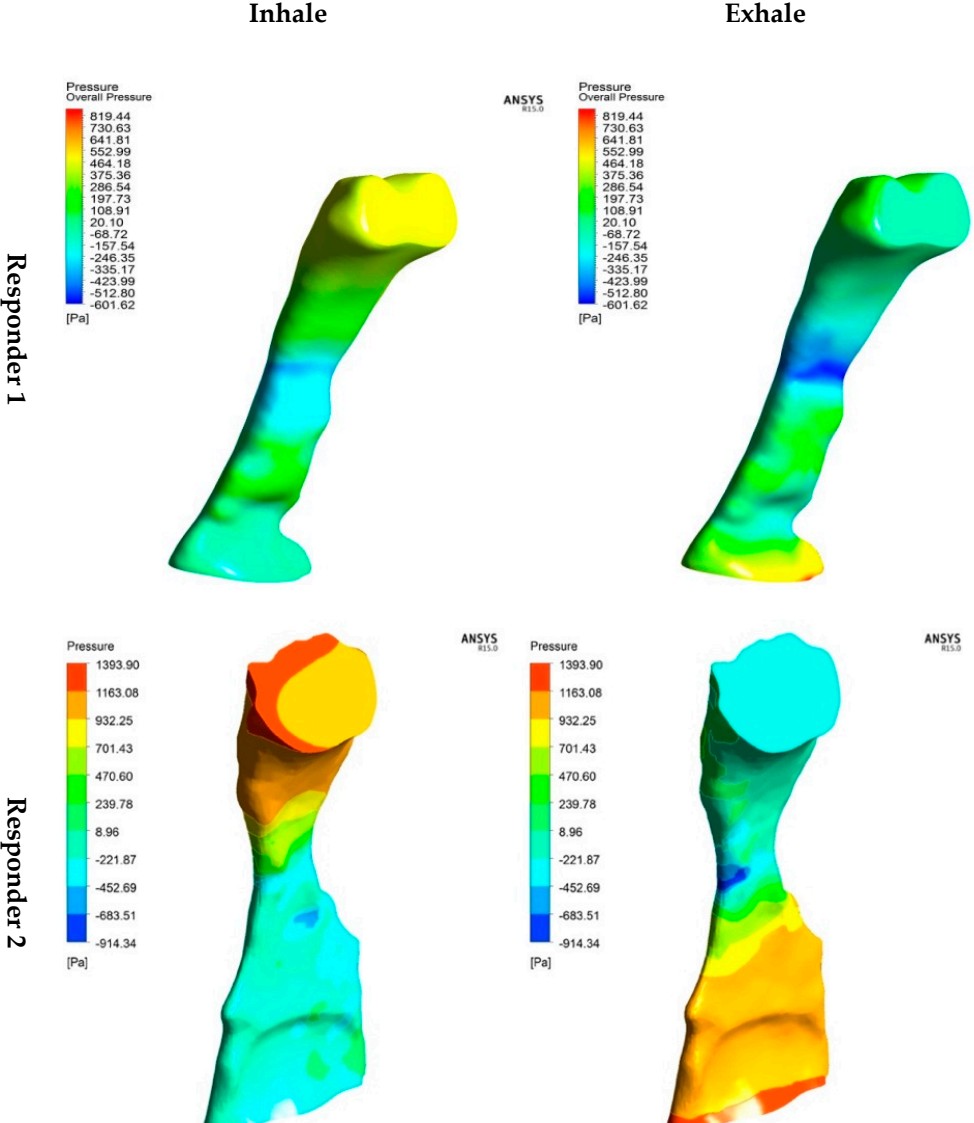

**Figure 6.** Contour of the static pressure exerted on the airway wall during inhaling and exhaling.

One such mechanism is activated in response to rapid changes in negative extrapharyngeal pressure, which causes the sub-atmosphere or suction pressure to increase. Indeed, advancing our comprehension of the neuroanatomy associated with the genioglossus negative pressure reflex and hypoglossal motor muscles has revealed the widespread presence of inhibitory inputs influencing the genioglossus. Nonetheless, while the responsiveness of the genioglossus muscles may be reduced during sleep compared to wakefulness, the muscles respond to sustained negative pressure.

### 3.3. Velocity

The velocity fluctuation exhibits a randomized direction, exerting a high magnitude that impacts the obstruction in the breathing area, as depicted in Figure 7. At the narrow cross-sectional breathing area, there is a notable low pressure. The obstructed regions primarily exhibit high wall shear stress, contributing to elevated turbulent energy near the wall. A recent investigation found that shear stress generates fluctuating velocity, leading to heightened turbulent energy in this region. The variations in velocity and turbulent flow near the wall induce vibrations, concurrently generating snoring and compromising the genioglossus function. Consequently, the severity of snoring correlates with turbulent energy. Over time, persistent snoring can diminish tissue strength in the pharyngeal region, potentially causing tissue collapse or laxity because of the gravitational effects when the patient is in a supine position. This scenario becomes a factor in the development of sleep apnea.

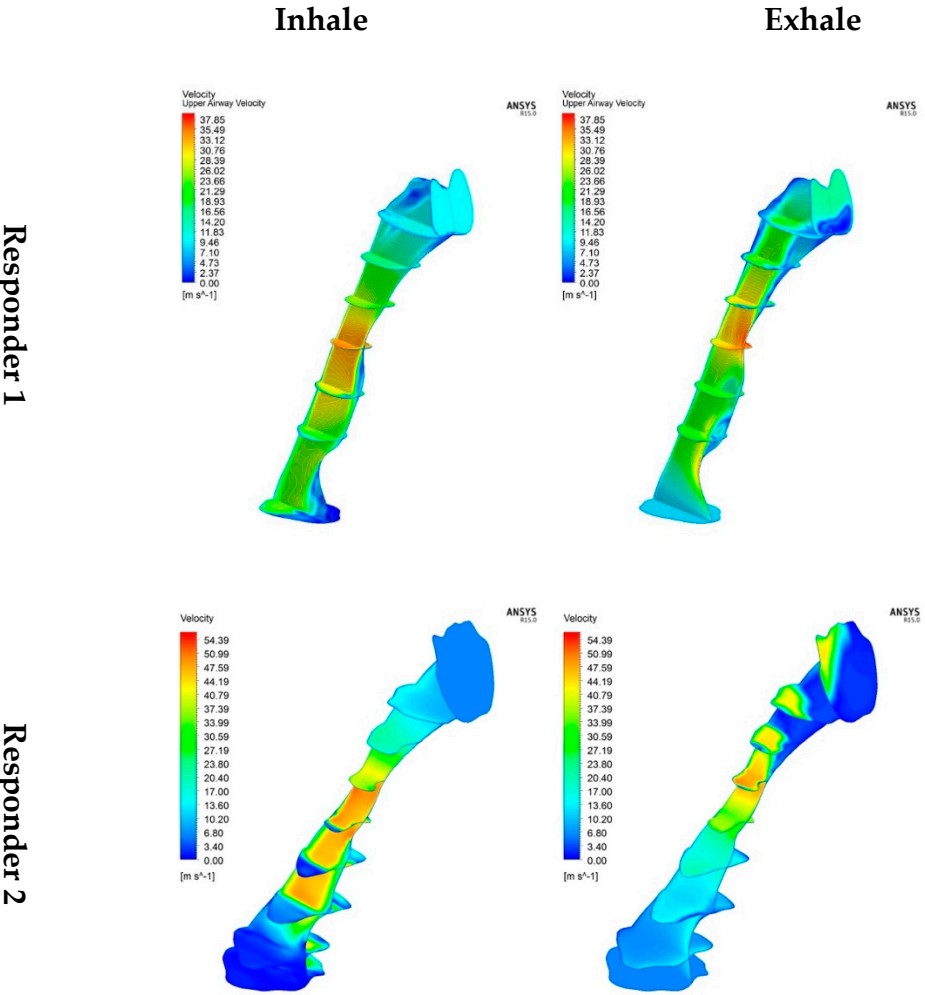

**Figure 7.** Contour of upper airway velocity magnitude on the airway wall during inhale and exhale breathing.

### 3.4. Turbulent Reynolds Number

Figure 7 shows the pattern of the turbulent Reynolds number for both responders and considers inhalation and exhalation breathing. It is noticed that after the critical cross-sectional area HUA, the flow pattern becomes turbulent. The maximum $Re_y$ for Responder 1 during inhalation and exhalation is 2398.44 and 1417.66, respectively. However, for Responder 2, the inhalation and exhalation $Re_y$ values are 3022.45 and 2134.13, respectively. The simulation results show that the $Re_y$ value of Responder 2 is higher than that of Responder 1. The difference between the two respondents is nearly 26.0% for inhaling and 50.5% for exhaling. This comparison reveals a reduced cross-sectional area in the airway (Responder 2), which led to increased airflow deterioration during inhalation and exhalation. The intensity of turbulent flow blocks the smooth airway during breathing, either during inhalation or exhalation. After the critical cross-sectional area, this turbulent condition will block the airway during breathing. The results revealed that the turbulent Reynolds number of both responders occurred after the area contraction during inhalation and exhalation breathing. The different cross-sectional areas between Responders 1 and 2 contribute to the different turbulence intensities after the critical cross-sectional area.

Nevertheless, during exhalation as shown in Figure 8, the occurrence of turbulence becomes more sophisticated. In the plane pattern after the inlet, the airflow already has higher turbulence than the inhalation breathing condition. Thus, the TKE was considered further to explain the airflow characteristics in the current study. It indicated that the expression of the turbulent Reynolds number is insufficient to measure or determine the severity of OSA as a new method.

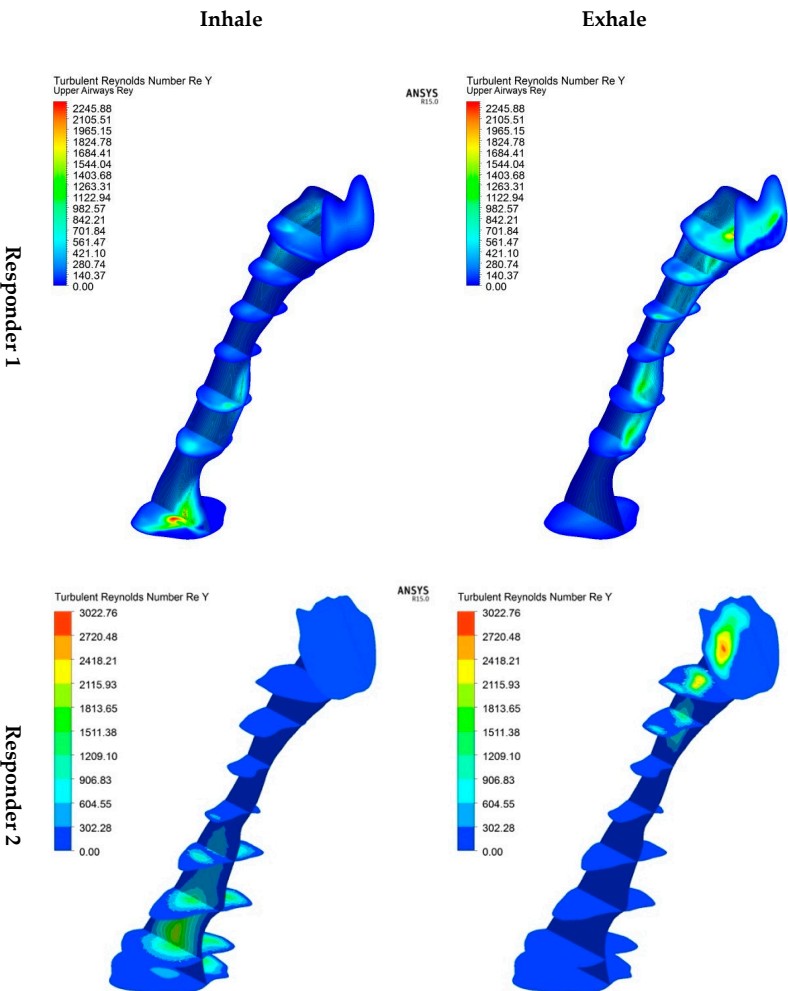

**Figure 8.** Contour of $Re_y$ at the center plane for inhale breathing and exhale breathing.

The correlation between the cross-sectional areas and turbulent Reynolds numbers can be measured, as expressed in Figure 9. The results revealed that the changes in turbulent Reynolds numbers of both responders occur after the area contraction during inhalation and exhalation breathing. This phenomenon corresponds to the velocity magnitude as discussed in Section 3.3. The different cross-sectional areas between Responders 1 and 2 contribute to the different turbulence intensities after the critical cross-sectional area. During inhalation, the turbulent behavior in airflow becomes more evident after the cross-sectional area of the upper airway for both responders. However, during exhalation, the occurrence of turbulence becomes more sophisticated. This phenomenon shows that in the pattern of the plane after the inlet, the airflow already has high turbulence compared to the inhalation breathing condition. This observation indicated that the expression of the turbulent Reynolds number is insufficient to measure or determine the severity of OSA as a new method. Thus, the turbulent kinetic energy was considered further to explain the airflow characteristics in the current study.

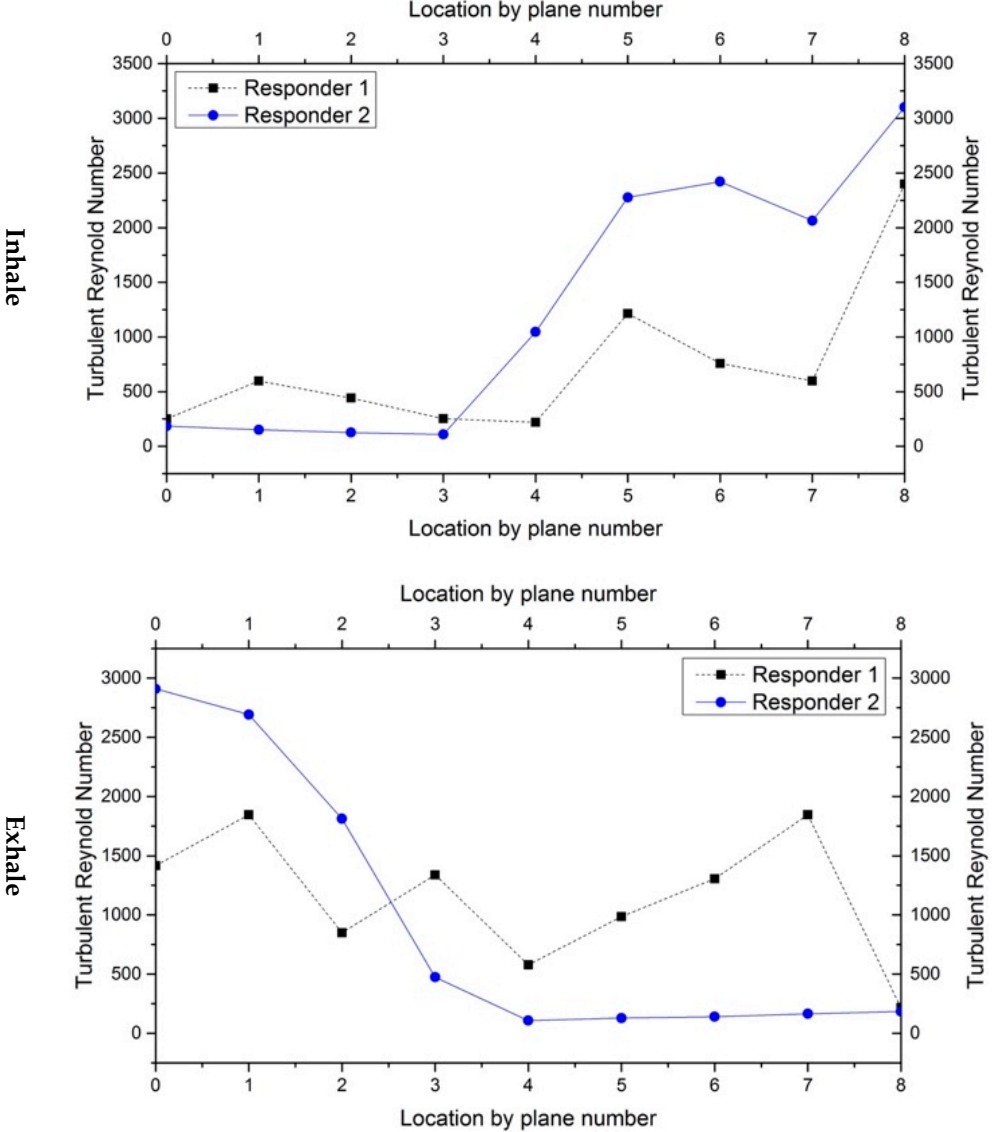

**Figure 9.** Turbulent Reynolds number trend for inhalation and exhalation for Responders 1 and 2.

### 3.5. Turbulent Kinetic Energy

Figure 10 illustrates the TKE contour resulting from the velocity fluctuation, as outlined in Equation (4). The turbulent Reynolds number ($Re_y$) was computed using Equation (6),

following the methodology established by Guangchu et al. [35]. In the airway, the predominant motion of airflow is attributed to energy in turbulent flow. Therefore, employing TKE is appropriate for describing vortices and quantifying turbulence levels. The outcomes reveal elevated TKE concentrated in the region following the narrow passage for both breathing conditions. Moreover, the formation of a jet stream was noted in the pharyngeal airway. The influence of this jet stream phenomenon on the wall probably resulted in the remodeling of velocity near the airway wall. The recirculation phenomenon induces the occurrence of TKE at high-velocity fluctuation.

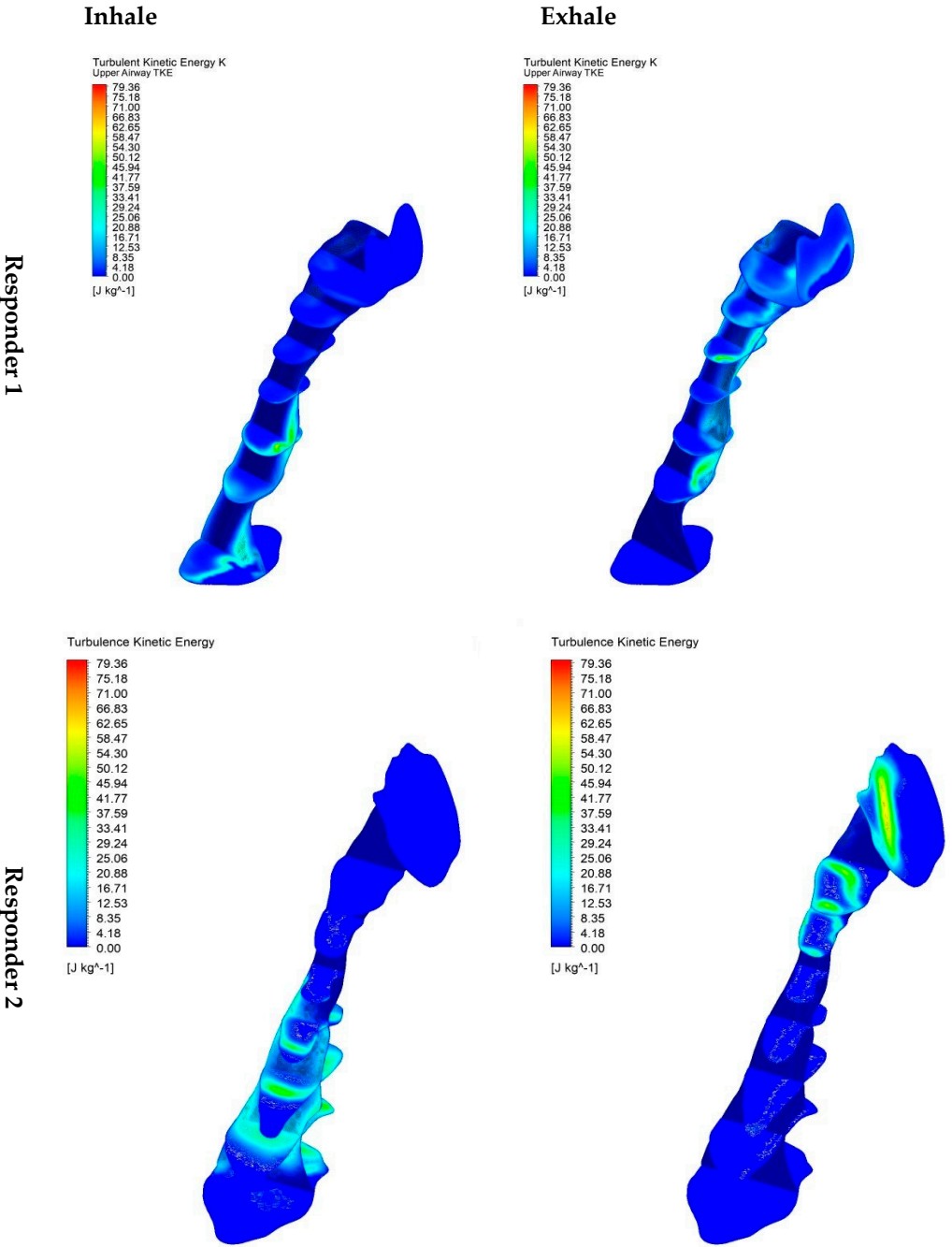

**Figure 10.** TKE contour of the upper airway at the center plane for inhale and exhale breathing conditions.

During inspiration, there is an increase in velocity from the inlet to the maximum value in the nasopharynx. In this stage, a high-velocity jet and increased turbulent kinetic

energy exist. Examining backflow and vortex downstream of the nasopharynx emphasizes the importance of the negative pressure gradient arising from the sudden expansion in the cross-sectional area. Throughout both inspiration and expiration, the model predicts localized areas of high TKE near the airway wall and the downstream jet. This phenomenon contributes to the relatively moderate pressure recovery in the jet expansion. The increased kinetic energy and dissipative energy loss upstream cause a decrease in pharyngeal luminal pressure during inspiration. Reduced luminal pressure reduces the cross-sectional area of the segment in hypotonic and extremely compliant pharynxes.

Figure 11 compares the turbulent kinetic energy of both responders during inhalation and exhalation breathing. The turbulent kinetic energy trends during breathing corresponded with the turbulent Reynolds number. In contrast, the sudden changes in TKE were spotted after the contraction area of the airway. The sudden increase and decrease in turbulent kinetic energy are found from Plane 3 to Plane 4 for both responders. The results show that the variations from Plane 3 to Plane 4 of Responder 2 are more than those of Responder 1. This situation is attributed to the critical cross-sectional area of Responder 2 being smaller than Responder 1. A smaller cross-sectional area showed variations in inhalation and exhalation before and after the narrowed area. Thus, it revealed that the critical cross-sectional area of the airway significantly affects the airflow characteristics of breathing, leading to the obstruction of sleep apnea.

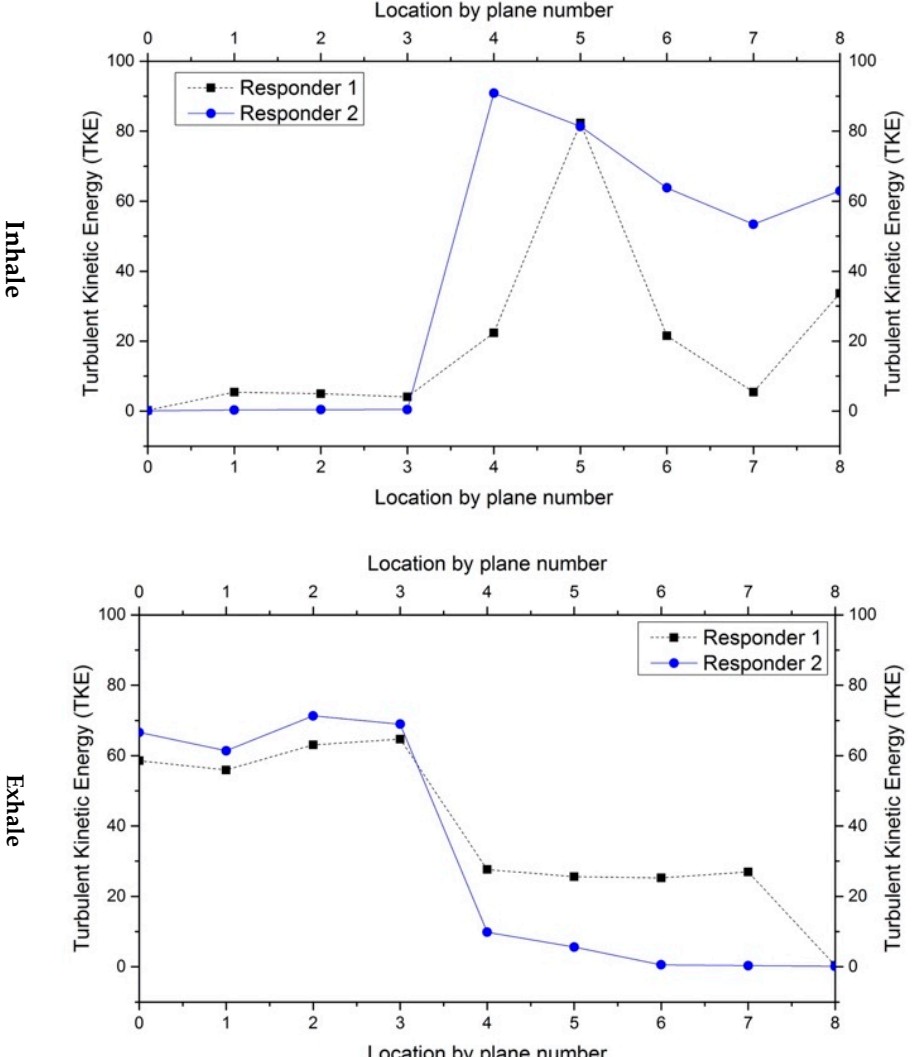

**Figure 11.** Turbulent kinetic energy trend for inhalation and exhalation for Responders 1 and 2.

The correlation between maximum pharyngeal TKE and cross-sectional area can be shown in relation to the OSA patient (Responders 1 and 2), respectively. The difference in cross-sectional areas was estimated at 29.47%. Reducing the cross-sectional area may increase TKE by almost 10.28% during inspiration and 10.18% for expiration (Table 2). The TKE increase was several-fold higher in the HUA after the critical cross-sectional area than in other segments in the control case.

**Table 2.** Comparison table between Responder 1 and Responder 2 during inhale and exhale breathing.

|  |  | Responder 1 | Responder 2 | Diff (%) |
|---|---|---|---|---|
| Cross-sectional area | | 7.68 mm$^2$ | 5.42 mm$^2$ | 29.47% |
| TKE | Inhale | 82.37 | 90.84 | 10.28% |
|  | Exhale | 64.72 | 71.31 | 10.18% |

## 4. Conclusions

The CFD method successfully simulates the flow characteristic during inhale and exhale breathing of OSA patients. The breathing condition has been considered in the simulations to study the impact of the cross-sectional area with different OSA patients. This study involved creating a 3D model of the pharyngeal airway using DICOM images and applying a solver verification to ensure the reliability of the computational model. ANSYS Fluent was utilized to set flow conditions, generate mesh, and conduct simulations, including a grid sensitivity test. It emphasized the significance of turbulent kinetic energy (TKE) in assessing obstructive sleep apnea (OSA) severity, highlighting its association with airflow characteristics. The research uncovered a substantial link between maximum pharyngeal TKE and cross-sectional areas in OSA patients, revealing that reduced cross-sectional areas could significantly increase TKE during inspiration and expiration. Decreasing the cross-sectional area could lead to a nearly 10.28% increase in TKE during inspiration and a 10.18% increase during expiration. This finding underscores the crucial role of TKE in understanding and evaluating OSA. Additionally, visualizing turbulent airflow patterns in individuals with pharyngeal airway irregularities provided valuable insights into potential factors contributing to OSA severity. These findings demonstrate how modeling can establish a specific patient's cross-sectional area and the underlying cause of hypopharyngeal airway (HUA) collapse. Thus, this information can be helpful to identify the most suitable treatment solution for OSA. The simulation results contribute to the profound understanding of the airflow patterns in the upper airway of OSA patients and facilitate individualizing OSA treatment based on underlying airflow mechanisms.

**Author Contributions:** W.M.F. designed the research, reviewed and revised the manuscript, approved the final manuscript as submitted, and obtained funding for the project. C.Y.K. analyzed and discussed the results and drafted and revised the manuscript. S.S. collected the data and carried out the research. M.H.M.H., M.A., M.N.M. and A.H.M.H. reviewed and revised the manuscript and approved the final manuscript as submitted. All authors have read and agreed to the published version of the manuscript.

**Funding:** This study was funded by the Ministry of Higher Education Malaysia (Ref. No: FRGS/1/2020/TK0/UNIMAP/03/26).

**Data Availability Statement:** The datasets used and analyzed during the current study are available from the corresponding author upon reasonable request.

**Conflicts of Interest:** The authors declare no competing interests.

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
