# Peer review of "Computational Fluid Dynamics Analysis of Varied Cross-Sectional Areas in Sleep Apnea Individuals across Diverse Situations"

_computation, doi:10.3390/computation12010016_

Round 1

Reviewer 1 Report

Comments and Suggestions for Authors

1.     Abstract section is very generic. Also should include the problem statement and some conclusive remarks

2.     Overall the paper is interesting and will catch the reader attention, however, I am unable to find the novelty in the article. Already, a lot of research is available on the said subject discussing the different cross-section based on gender age and weight. Also, the TKE model has been used during number of related studies.

3.     Author should include the table discussing the different corss-section area with different turbulent model. Also, the shortcoming in each analysis.

4.     No information is provided related to the quality of the mesh. Only number of nodes/ elements, mesh size and quality are not sufficient to declare it as better mesh. Author should clarify the y+ requirement for each mesh. More information should be included related to mesh optimization and the nature of variation in mesh.

5.     Why author not prefer to use the RAN-ke model?

6.     Conclusion section is not well written. It look like a book paragraph. No quantitative conclusion has been drawn.

Comments on the Quality of English Language

Moderate editing of English language required

Author Response

I am herewith enclosing the revised manuscript titled “CFD ANALYSIS OF DIFFERENT CROSS-SECTIONAL AREAS OF SLEEP APNEA PATIENTS UNDER VARIOUS CONDITIONS” been invited by Prof. Dr. Ali Cemal Benim for possible publication in Special Issue "10th Anniversary of Computation—Computational Heat and Mass Transfer (ICCHMT 2023)”. It is indicated in the manuscript by two colors. The red indicates an improvement based on a reviewer's feedback, whereas the blue indicates an editor's proposal for a similarity enhancement. But the blue color also denotes certain enhancement suggestions made by the reviewer. Please see the attachment for a detailed explanation.

Reviewer 2 Report

Comments and Suggestions for Authors

This research investigated obstructive sleep apnea (OSA), a prevalent medical condition affecting a significant portion of the population. The primary goal of the research is to provide a deeper understanding of how the cross-sectional area affects airflow characteristics in the upper airway, potentially contributing to the development of a new assessment method utilizing TKE for doctors studying OSA. The study focused on analyzing inhale and exhale breathing airflow parameters in OSA patients. The authors created a 3D airway model using advanced software. To simulate upper airway airflow, the researchers employed a steady-state RANS approach and an SST turbulence model. The simulations indicate that the cross-sectional area of the airway significantly influences velocity, Reynolds Number, and TKE, which may be used to assess the severity of OSA.

Please see detailed comments below.

Abstract

1. It suggests reserving the sequence between 2nd and 3rd sentences in Abstract. It might be better to mention how the virtual airway geometry was constructed, and then describe the modeling methods.

Introduction

2. OSA was defined several times. Please use abbreviation after the first use.

3. Line 64, please define TKE in the text.

4. Line 79, what is "airway pressures", how is airway pressure measured, locations? Is it an averaged value during breathing?

5. Line 87-89, this sentence is very ambiguous. Please revise and make the statement clear.

6. Line 95-98, please provide proper references for these statements.

Materials and Methods

7. Line 116, please define HUA before using abbreviation.

8. Was the airway scanned during inhalation or exhalation? Did the authors consider the airway wall deformation during breathing?

9. Please revise the passage to ensure consistency in tense.

10. Line 198-199, this statement is questionable. Previous work has shown that the glottal motion has impact on the flow pattern in the upper airway.

Zhao, J., Feng, Y., Fromen, C. (2020). Glottis Motion Effects on the Inhaled Particle Transport and Deposition in a Subject-Specific Mouth-to-Trachea Model: An in silico Study. Computers in Biology and Medicine. 116, ID: 103532.

11. Please indicate that the gravitational force was neglected in the CFD simulation.

12. Equation (4) is the definition of TKE in theory, however, it is not how TKE is calculated in RANS k-omega SST model. Please clarify this in the method.

13. For grid sensitivity study, please provide the data for both HUA geometries.

14. For KE, was the averaged velocity used or real velocity?

Results and Discussion

15. Was there any reversed flow at the outlet during CFD simulation?

16. For Figure 7, it suggests using term "velocity magnitude" instead of "velocity".

17. How were the turbulent intensity and hydraulic diameter at the inlet determined in this study, especially for exhalation phase?

Comments on the Quality of English Language

Please see above comments.

Author Response

(The authors gave the same response as above.)

Round 2

Reviewer 1 Report

Comments and Suggestions for Authors

Author addressed the reviewer comments and article is suitable for publication.

Comments on the Quality of English Language

Moderate English editing required

Author Response

To Reviewer,

Thank you for your input on how to improve the article.

Reviewer 2 Report

Comments and Suggestions for Authors

All my previous comments and concerns have been excellently addressed in the revised manuscript. Only one minor suggestion for the choice of the word: line 243, it is better to say "model/calculated the TKE" instead of "define the TKE".

This manuscript could potentially be interesting to readers with either clinical or modeling background. Therefore, the publication of this paper in Computation is recommended.

Author Response

To Reviewer,

I am hereby enclosing the 2nd revised manuscript titled 'CFD ANALYSIS OF DIFFERENT CROSS-SECTIONAL AREAS OF SLEEP APNEA PATIENTS UNDER VARIOUS CONDITIONS.' The comments have already been corrected accordingly, including some minor edits to the English language.

Thank you for your input on how to improve the article.
